



# Interrogating process deficiencies in large-scale hydrologic models with interpretable machine learning

Admin Husic[1], John Hammond[2], Adam N. Price[3], Joshua K. Roundy[4]

[1]Department of Civil and Environmental Engineering, Virginia Tech, Blacksburg, Virginia, USA
5  [2]U.S. Geological Survey, Maryland-Delaware-D.C. Water Science Center, Catonsville, Maryland, USA
[3]USDA Forest Service, Pacific Northwest Research Station, La Grande, Oregon, USA
[4]Department of Civil, Environmental and Architectural Engineering, University of Kansas, Lawrence, Kansas, USA

*Correspondence to*: Admin Husic (husic@vt.edu)

10  **Abstract.** Large-scale hydrologic models are increasingly being developed for operational use in the forecasting and planning of water resources. However, the predictive strength of such models depends on how well they resolve various functions of catchment hydrology, which are influenced by gradients in climate, topography, soils, and land use. Most assessments of these hydrologic models has been limited to traditional statistical approaches. The rise of machine learning techniques can provide novel insights into identifying process deficiencies in large-scale hydrologic models. In this study, 15  we train a random forest model to predict the Kling-Gupta Efficiency (KGE) of National Water Model (NWM) and National Hydrologic Model (NHM) predictions for 4,383 streamgages across the conterminous United States. Thereafter, we explain the local and global controls that 48 catchment attributes exert on KGE prediction using interpretable Shapley values. Overall, we find that soil water content is the most impactful feature controlling successful model performance, suggesting that soil water storage is difficult for hydrologic models to resolve, particularly for arid locations. We identify non-linear 20  thresholds beyond which predictive performance decreases for NWM and NHM. For example, soil water content less than 210 mm, precipitation less than 900 mm/yr, road density greater than 5 km/km$^2$, and lake area percent greater than 10% contributed to lower KGE values. These results suggest that improvements in how these influential processes are represented could result in the largest increases in predictive performance of NWM and NHM. This study demonstrates the utility of interrogating process-based models using data-driven techniques, which has broad applicability and potential for improving 25  the next generation of large-scale hydrologic models.

## 1 Introduction

Large-scale hydrologic models are important tools for understanding and forecasting the fluxes of water across the earth's surface to manage floods, droughts, and other hydrologic extremes (Brunner et al., 2021; Tijerina et al., 2021). Most often, these models convert meteorological inputs to streamflow predictions by parameterizing and calibrating internal 30  hydrological processes. Accurate simulation of internal processes is a grand challenge of hydrology (Blöschl et al., 2019)



because of the difficulty of resolving equifinality (Vrugt and Beven, 2018), scaling relationships (Savenije, 2018), epistemic uncertainties in hydrologic data (Beven, 2024), and spatial heterogeneity in watershed attributes (McDonnell et al., 2021). Accurate determination of model limitations is crucial for improving process representation in hydrologic models and, ultimately, the management of water resources.

35  The National Water Model (NWM) and the National Hydrologic Model (NWM) are two process-oriented, continental-scale hydrologic models designed for use in operational decision-making (Towler et al., 2023). The NWM framework applies the WRF-Hydro formulation, which includes representations for infiltration, evaporation, transpiration, overland flow, shallow subsurface flow, baseflow, channel routing, and passive reservoir routing, but not active reservoir management (Cosgrove et al., 2024). The NHM framework applies the Precipitation-Runoff Modeling System formulation,

40 which includes representation of evaporation, transpiration, runoff, infiltration, interflow, groundwater flow, and channel routing, but not reservoir operations, water withdrawals, or stream releases (Regan et al., 2019). Perhaps the major difference in the two modeling approaches is that the NWM has a focus on high-resolution (hourly) flood forecasting whereas the NHM is designed to assess general water availability at timescales from days to centuries (Towler et al., 2023). The NWM and NHM have variable success for streamflow prediction (Tijerina et al., 2021), which depends on differences between sites

45 in catchment-scale climate, land use, and physiographic regimes.

  The sensitivity of process-based hydrologic models to certain catchment attributes and parameters has been interrogated using well-established tools, such as sensitivity analyses (Song et al., 2015). These approaches work by exploring the range of values that model parameters may take and recording the net impact on model performance (Mai, 2023). Mai et al. (2022) showed in a recent large-scale sensitivity analysis across North America that functional relationships

50 could be derived between hydrologic processes and physiographic catchment characteristics. In those studies, the authors excluded poor performing sites from their sensitivity analyses (NSE < 0.50); however, poor performing sites may have the greatest potential for identifying sensitive processes and improving their representation in hydrologic models. Thus, there is a need to understand the characteristics of catchments that lead to poor performance to improve confidence in operational decision making in diverse settings.

55  Machine learning has transformed the field of hydrology in recent years, providing improved predictive capabilities (Kratzert et al., 2018). These data-driven approaches have highlighted that large-scale hydrological datasets contain more information in them than is explained by our existing theories and perceptions (Nearing et al., 2021). To this end, explainable or interpretable artificial intelligence (AI) methods can be leveraged to bridge the gap between data driven understanding (provided by machine learning models) and process based understanding (contained within physically based

60 models) (Park et al., 2022). Numerous explainable AI methods have been developed, including Partial Dependence Plots (PDP; Friedman, 2001), Local Interpretable Model-Agnostic Explainers (LIME; Ribeiro et al., 2016), and Shapley Additive Explanations (SHAP; Lundberg et al., 2020). Thus, there is an opportunity to apply data-driven, explainable AI approaches to identify sensitive processes in physically based hydrologic models.



Explainable AI can complement and enhance traditional approaches like Sobol's sensitivity analysis (Nossent et al., 2011) and the Generalized Likelihood Uncertainty Estimation (GLUE; Blasone et al., 2008) method in hydrologic modeling by providing deeper insights and interpretability. Explainable AI techniques, such as SHAP values, enhance traditional sensitivity analyses like Sobol by providing interpretative insights into how parameter changes influence model predictions. Similarly, explainable AI complements the GLUE methodology by visualizing the impact of uncertain parameters on output variability and offering local explanations for specific predictions. However, caution is necessary when inferring AI results 70 because they typically only imply direct or indirect relations and may not represent causal linkages (Heskes et al., 2020).

This paper aims to interrogate large-scale hydrologic model performance with machine learning tools to identify which processes may be inadequately represented in physically based models. Thus, the questions we address are: what catchment attributes can be used to predict poor model performance, and are certain dominant hydrological processes associated with these catchment attributes? To answer these questions, we built a random forest machine learning model to 75 predict KGE values for NWM and NHM predictions at over 4,000 basins. Thereafter, model predictions were interpreted using Shapley values, which highlight the physiographic and hydrologic controls of process-based model performance (Fig. 1). This work aims to inform how the next generation of large-scale hydrologic models can be improved for the responsible stewardship of water resources into an uncertain future.

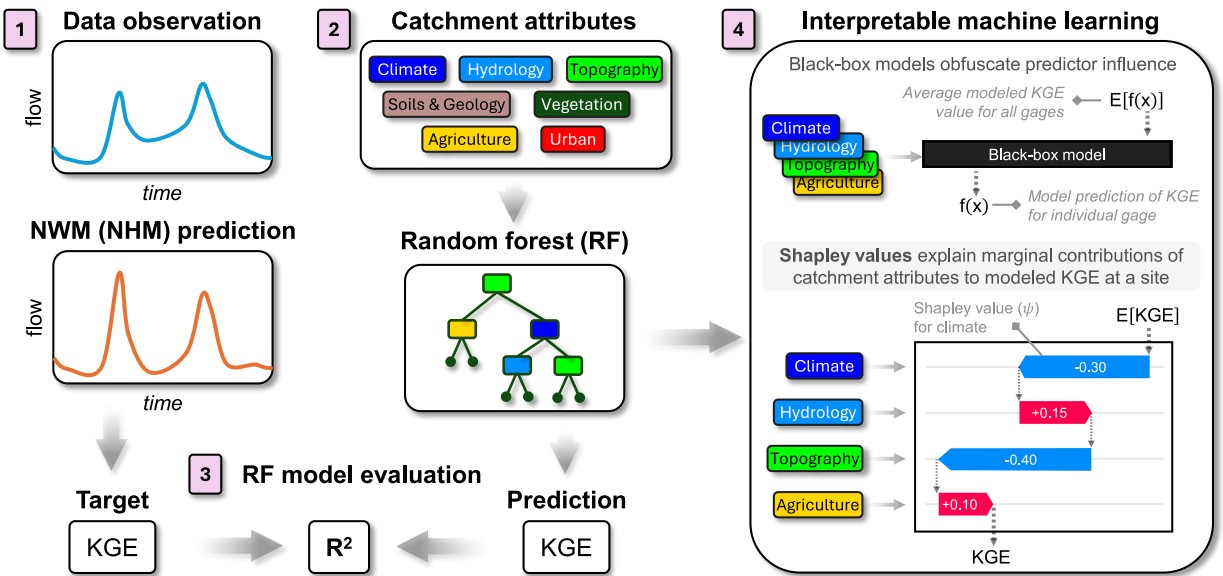

**Figure 1:** Flow diagram showing the application of interpretable machine learning in this study. (1) Data observations and National Water Model (NWM) or National Hydrologic Model (NHM) predictions are used to generate a target Kling–Gupta efficiency (KGE) for each site. (2) Catchment attributes are input to a Random Forest (RF) model to predict KGE for each site. (3) The RF model is evaluated by comparing the predicted KGE to the target KGE. (4) Shapley values ($\psi$) are used to explain the marginal contributions of catchment attributes that distinguish KGE prediction at a particular site, f(x), from the average modeled KGE for all sites, E[(f(x)]. In the given 85 example, the values of the climate and topography attributes at this individual gage lower the predicted KGE ($-\psi$), whereas the values of the hydrology and agriculture attributes increase the predicted KGE ($+\psi$) by the RF. $R^2$ = coefficient of determination.



## 2 Methods

### 2.1 The National Water and National Hydrologic Models

We retrieve daily streamflow observations and predictions for gaged locations (sites) for the NWM version 2.1 and
NHM version 1.0 from existing repositories (Johnson et al., 2023a; Regan et al., 2019). A total of 4,614 basins that span the
contiguous US (CONUS) are included in our analysis (U.S. Geological Survey, 2024). The date range of flow observations
and predictions is from water years 1984 to 2016. The accuracies of NWM (Fig. 2) and NHM (Fig. S1) predictions are
particularly sensitive to aridity. Model performance was assessed at each site using the Kling-Gupta Efficiency (KGE), a
common evaluation metric for hydrologic modeling (Gupta et al., 2009). The KGE is calculated as

$$KGE = 1 - \sqrt{(\alpha - 1)^2 + (\rho - 1)^2 + (\beta - 1)^2} \tag{1}$$

where $\rho$ is Pearson correlation coefficient, and $\alpha$ and $\beta$ are the ratios of the standard deviation and the mean,
respectively, of model predictions to data observations.

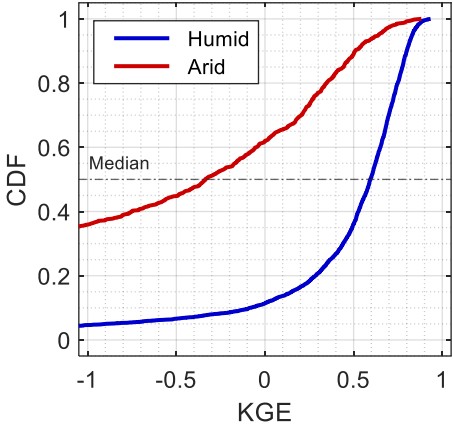

**Figure 2:** Cumulative distribution function (CDF) of National Water Model (NWM) performance for humid (PET/P <1, n = 3,827) and
arid (PET/P >1, n = 787) sites as assessed by the Kling–Gupta efficiency (KGE) evaluation metric.

### 2.2 Random Forest Model

A regression tree is a supervised learning approach that can predict continuous values and capture non-linear trends
in a dataset (De'Ath and Fabricius, 2000). A random forest model creates an ensemble of regression trees to mitigate the
potential of overfitting to a single regression tree (Ho, 1998). In this study, we train 1,000 regression trees to predict KGE.
The predictor variables (termed "features") used to train the model are 48 catchment attributes, which were aggregated based
on their likelihood to impact hydrology. The features are derived from BasinATLAS (Linke et al., 2019) and incorporate
wide ranges of climate, hydrology, topography, soils & geology, natural vegetation, agriculture, and urban land use. The
names and descriptions of the 48 predictors can be found in Table S1, and the spatial variations of the 48 predictors across
the CONUS are shown in Fig. S2. One influential predictor – soil water content – is defined as soil stress, or the annual soil



water available for evapotranspiration (Trabucco and Zomer, 2010). We represent soil water content as an equation equal to the long-term effective precipitation minus the sum of actual evapotranspiration and runoff.

Training of the random forest was done with "in-the-bag" and "out-of-bag" splits. Individual trees are grown from an "in-the-bag" bootstrap of the observation dataset. "Out-of-bag" observations not included in the bootstrap are used for model validation. The models were trained using the mean squared error objective function. The coefficient of determination

($R^2$) was calculated to assess predictive performance of the random forest (Pearson, 1901). Extreme values (outliers) can distort the utility of a predictive and interpretable model (Liu et al., 2018). Because the KGE metric has a small upper bound (+1) and an infinite lower bound ($-\infty$), a small subset of very negative values can dominate model inferences. The lowest KGE value for a gaged location in the NWM dataset is -302.8, whereas the 5th percentile of KGE values -2.7. The performance at both sites would be considered "unacceptable"; thus, including extreme negative values negatively affects

model predictability without providing much additional insight beyond that given by other underperforming sites. To address the disproportionate influence of a small subset of values, we consider the 5% of sites with the most negative KGE values as outliers, reducing our dataset from 4,614 to 4,383 sites. Random forest model analyses and development were performed using the treebagger function in MATLAB 2023 (MathWorks, 2023).

### 2.3 Shapley Values

The Shapley value approach is a model-agnostic, explainable AI method that attributes each feature an importance value for a prediction, indicating the marginal benefit that the inclusion of a particular feature provides to the overall prediction (Lundberg et al., 2020; Lundberg and Lee, 2017). The Shapley value has the same units as those of the prediction. Further, the Shapley value is also the only distribution of gain among features (e.g., predictor variables) that maximizes four properties: (1) efficiency, (2) symmetry, (3) linearity, and (4) null player (Shapley, 1953). Thus, while other model

explanation techniques exist, e.g., LIME (Ribeiro et al., 2016), they violate one or more of these properties.

The Shapley value ($\psi$) of the i-th feature (catchment attribute) for the query point x (KGE) can be calculated by the characteristic value function (v) as:

$$\psi_i(v_x) = \frac{1}{M} \sum_{S \subseteq M \setminus \{i\}} \frac{|S|! \, (M - |S| - 1)!}{(M - 1)!} \, [v_x(S \cup \{i\}) - v_x(S)] \qquad (2)$$

where M is the number of features, *M* is the set of all features, S is a set or coalition of features, |S| is the number of elements in the coalition, $v_x(S)$ is the value function of the features in the coalition for the query point x (Shapley, 1953). The

value of $v_x$ (S) represents the "worth" or the expected contribution of the features in S to the cooperative prediction for the query point x. Leveraging the additive nature of Shapley values, we calculate them for each observation for all trees in the random forest and then average respective feature results across trees for a more robust statistic. All Shapley value analyses were performed using the TreeSHAP function in MATLAB 2023 (Lundberg et al., 2020; MathWorks, 2023).

Although the full range of Shapley values for the 48 catchment attribute features are informative, we focus on the

most impactful feature negatively affecting model performance at each site. The most impactful feature is defined as the one



having the lowest Shapley value (min $\psi$) at a site, i.e., providing the most negative marginal contribution to KGE prediction. We relate our Shapley value feature importance to the spatial distributions of the Ecological Regions of North America (Omernik, 1987), where watersheds were assigned to Ecoregions based on the greatest area of an ecoregion in the watershed. Ecoregions are defined by "perceived patterns of a combination of causal and integrative factors including land use, land-
surface form, potential natural vegetation, and soils. We consider Level-I and Level-II ecoregions, which will be identified in figures using the following superscripts: [1]Atlantic Highlands, [2]Mixed Wood Shield, [3]Ozark, Ouachita-Appalachian Forests, [4]Mixed Wood Plains, [5]Central USA Plains, [6]Southeastern USA Plains, [7]Mississippi Alluvial and Southeast USA Coastal Plains, [8]Everglades, [9]Temperate Prairies, [10]West-Central Semi-Arid Prairies, [11]South Central Semi-Arid Prairies, [12]Texas-Louisiana Coastal Plain, [13]Tamaulipas-Texas Semi-Arid Plain, [14]Cold Deserts, [15]Warm Deserts, [16]Western Sierra Madre
Piedmont, [17]Upper Gila Mountains, [18]Western Cordillera, [19]Marine West Coast Forest, and [20]Mediterranean California.

## 3 Results

Because general results for both the NWM and NHM were broadly similar, we focus the main text discussion on the NWM and note instances where the two models differ (detailed results from NHM analysis can be found in the Supplement). $R^2$ values for the training and testing predictions of KGE for the random forest model were 0.86 (0.86) and
0.47 (0.43), respectively, for the NWM (NHM). The criteria for acceptability of $R^2$ varies with the complexity of a dataset (Legates and McCabe, 1999), and we consider a model that explains 47% of the variance encoded in the KGE metric for 4,383 gages as acceptable to function as a surrogate for predicting NWM performance. We apply explainable AI techniques to this random forest model to understand how catchment attributes influence KGE values of streamflow for the NWM and NHM.

We investigated the local structure of Shapley values ($\psi$) for three demonstration sites (Fig. 3). We report how the Shapley values explain random forest model predictions of KGE, but these explanations may not be a result of direct causality. The directionality and extent of influence by each predictor is indicated by the magnitude and sign of the predictor's Shapley value ($\pm\psi$). Each waterfall plot shows how Shapley values ($\psi$) of features help to distinguish one site, f(x), from the mean of all sites, E[f(x)]. These three sites were selected to demonstrate various catchment controls, such as
climate at Tucannon River, WA; hydrology at Seboeis River, ME; and soils & geology at Timpas Creek, CO. At Tucannon River, the relatively high values of actual evapotranspiration and aridity index at the site cause a decrease (-$\psi$) in the prediction of KGE at that site. At Seboeis River, the large lake area percentage causes a decrease (-$\psi$) in KGE prediction, but the high soil water content causes an increase (+$\psi$) in KGE prediction. At the final site, Timpas Creek, the most influential feature is the low soil water content, which has a considerable negative contribution (-$\psi$) to KGE prediction. With an
understanding of the local structure of Shapely values, we proceed to a global perspective by assessing the aggregate results of all 4,383 sites.







**Figure 3:** Local structure of Kling–Gupta efficiency (KGE) prediction for the National Water Model (NWM) as illustrated by Shapley value ($\psi$) waterfall plots at three demonstration sites, indicated by U.S. Geological Survey station numbers associated with streamgages and 2-letter state abbreviations. Each plot begins with the expected value of the model prediction for all sites, E[f(x)], which undergoes marginal alteration (±$\psi$) by each of the 48 predictor features. The final model prediction, f(x), is equal to E[f(x)] plus the cumulative sum of all marginal contributions. Undeveloped Vegetation is abbreviated as Und. Veg.





The global structure of Shapley values ($\psi$) for six important catchment attributes is shown (Fig. 4): soil water
content, snow cover maximum, road density, precipitation, lake area, and irrigated area. The marginal contribution of the soil
water content variable ($\psi_{\text{soil water content}}$) is positive ($+\psi$) in areas with high soil water content (east of the 98$^{\text{th}}$ meridian and in
the Pacific Northwest) and negative ($-\psi$) in areas with lower soil water content (Great Plains, Intermountain West, and
California). The Shapley dependence plot identifies 210 mm soil water content as a threshold from when $\psi_{\text{soil water content}}$
increases ($+\psi$) versus decreases ($-\psi$) the prediction of KGE. The $\psi_{\text{snow cover max.}}$ values are positive in the Rocky Mountains
and the upper Midwest. Snow cover maximum has little effect on KGE predictions until a threshold of 40% is exceeded, at
which point maximum snow coverage improves KGE prediction. The $\psi_{\text{road density}}$ values are negative in urban centers, when
road density exceeds 5 km/km$^2$, suggesting high road density decreases accuracies of model predictions. Otherwise, the
presence of roadways has little impact on KGE predictions at lower road densities. A threshold of 900 mm/yr in precipitation
emerges; precipitation values lower than this threshold lower KGE ($-\psi_{\text{precipitation}}$) and values greater than this threshold
increase KGE ($+\psi_{\text{precipitation}}$). The $\psi_{\text{lake area}}$ values are generally close to zero except for when lakes constitute a substantial
portion of a watershed ($> 10\%$), such as in Minnesota and Wisconsin and the Northeast Region. For $\psi_{\text{irrigated area}}$, watersheds
with less than 3% irrigated area are unaffected by the variable, but beyond a threshold of around 10%, the presence of
irrigation decreases KGE predictions.

Shapley value swarm charts show the directionality and magnitude of feature importance for all 48 predictors (Fig.
5). Globally, the most impactful features (greatest $\overline{|\psi|}$) for KGE prediction are $\psi_{\text{soil water content}}$, $\psi_{\text{aridity index}}$, $\psi_{\text{actual ET}}$, and
$\psi_{\text{precipitation}}$. Regarding directionality, higher catchment-scale values of soil water content, aridity index, actual ET, and
precipitation increase KGE prediction ($+\psi$) whereas smaller values decrease KGE prediction ($-\psi$). Although these are
globally the most influential variables, they are not necessarily the most influential at each individual site. We plot the spatial
distribution of the most impactful feature group leading to poor KGE scores at each site, that is the predictor group having
the greatest negative Shapley value (min $\psi$) at a site. The count of most impactful features groups at individual sites were
climate (n = 761), hydrology (n = 1,290), and soils and geology (n = 1,447). Soils and geology features, most frequently low
soil water contents, reduced KGE most often in the Great Plains and Intermountain West. Hydrology features, typically large
values of lake and reservoir storage, reduce modeled KGE in the Midwest. Climate features did not have strong spatial
coherence. Next, we assess the distribution of KGE values grouped by most impactful feature (Fig. 6). For the NWM, sites
where the most impactful features were soils & geology as well as urban land use had the lowest median KGE values. The
results for NHM were similar to NWM except that areas controlled by climate have lower median KGE values for NHM
than NWM.

We map the spatial linkage between ecological regions in the US and the influential features controlling KGE
scores at sites contained within these regions (Fig. 7). The ecoregions containing the most streamgages are Eastern
Temperate Forest, Great Plains, Northwestern Forested Mountains, and North American Deserts. Streams in the Eastern
Temperate Forest ecoregions are most frequently influenced by, in decreasing order, hydrology, climate, urban, and soils &





geology features. For the Great Plains, the most frequent controlling features are soils & geology, followed distantly by hydrology. The Northwestern Forested Mountains are influenced by soils & geology, climate, hydrology, and topography. Lastly, the North American Desert streams are controlled almost exclusively by soils & geology features.


**Figure 4:** Spatial distribution of Shapley values ($\psi$) for selected influential features and their impact on Kling–Gupta efficiency (KGE) prediction for the National Water Model (NWM). The partial dependence plot of each feature is shown. Features value distributions are
represented with a heatmap. A moving average of feature values is indicated by a line to show general trends.





**Figure 5:** (a) Map of Kling–Gupta efficiency (KGE) for the National Water Model. (b) Map and histogram of the most impactful feature causing poor model performance at each site, i.e., the predictor group having the greatest negative Shapley value ($\psi$) at a site. (c) Swarm chart of Shapley values for KGE prediction showing feature importance for 48 predictors. The staircase plot on the right axis indicates the mean absolute Shapley value $\overline{|\psi|}$) of all observations for a predictor.



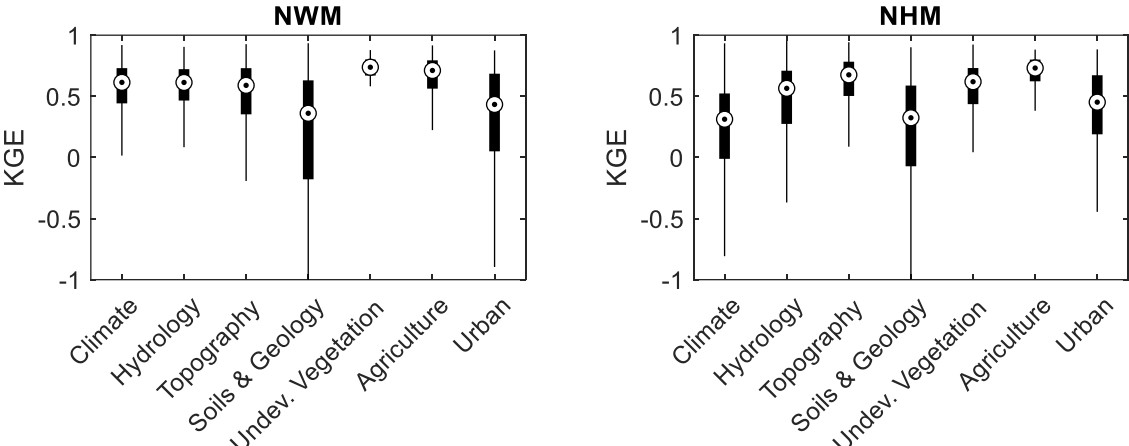

**Figure 6:** Kling–Gupta efficiency (KGE) performance grouped by the most important variable at each site as identified by Shapley values for the National Water Model (NWM) and National Hydrologic Model (NHM).


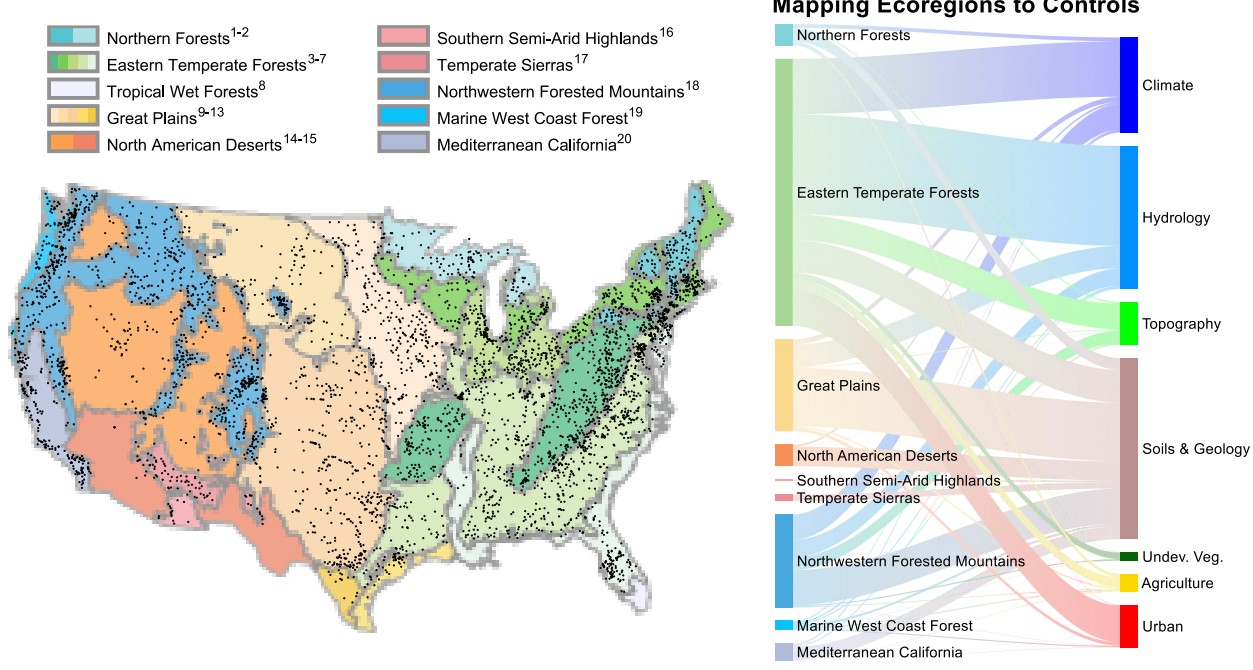

**Figure 7:** Map of study stream gages (black markers) and the Ecological Regions of North America (as defined in Omernik, 1987). Sankey diagram showing the pairing of ecoregions and impactful feature classes for the National Water Model (NWM) for the Kling–
Gupta efficiency (KGE) evaluation metric. Superscripts in ecoregion classifications are defined in Section 2.3.



## 4 Discussion

We investigate the relative importance of catchment attributes to streamflow model performance to diagnose deficiencies in how the hydrologic models represent physical processes. Compared to other parameter-based continental-scale sensitivity analyses (e.g., Mai et al., 2022), our approach provides a post-hoc assessment of model sensitivity. That is, perturbing the parameterization of the original modeling framework is not necessary to identify model sensitivities. Rather, sensitivities are deduced (learned) through the identification of the marginal contribution of predictor features to model performance. That is, our approach identifies how catchment attributes may impact the predictions of KGE at a site. The interpretable machine learning approach we present is flexible and model agnostic, meaning it can be applied to any modeling framework.

### 4.1 Model diagnostics with explainable AI

The Shapely value approach used in our study is able to make both local (Fig. 3) and global (Fig. 4) inferences from the same model. Shapley dependence plots allow us to infer the individual (marginal) contribution of a feature to the overall model as a function of the feature's magnitude. Other approaches, such as LIME (local) and PDP (global) can only calculate one or the other. Shapley values have also been shown to match human intuition more closely compared to LIME (Lundberg and Lee, 2017), providing confidence in the approach. Below, we highlight both local and global structures that emerge from our analysis and that allow for the interrogation of NWM and NHM model performance.

Local structures emerge whereby a few sensitive attributes can dominate the overall KGE prediction at a site (Fig. 3). This can manifest as a catchment attribute decreasing or increasing prediction accuracies (as measured by KGE) of NWM or NHM. For example, at an arid site on the Tucannon River (WA), the NWM performance is lower at this site than the nation-wide average of NWM for all sites because of high actual evapotranspiration and low precipitation conditions. Conversely, at Seboeis River (ME), the higher humidity and soil water content contributes to higher NWM prediction accuracy compared to the nation-wide average site. In some instances, multiple competing attributes offset their negative and positive contributions to KGE prediction. At the Seboeis River, the positive contribution to KGE from high soil water content is offset by the negative contribution of a large lake area percentage. Another way to interpret this would be that in the absence of lakes in the basin, the NWM would produce more accurate streamflow predictions at this site, i.e., a higher KGE. Therefore, although the model's representation of soil water content at this site increases streamflow prediction accuracy, the simulation of lake water storage (or lack thereof) is inhibiting streamflow prediction. Importantly, the Shapley value approach can also identify features that are not influential to KGE. For example, for all three sites investigated in Fig. 3, the natural vegetation and agricultural variables have limited influence on KGE. By elucidating the local structure of catchment controls on model performance, this approach allows for inference about which processes are not well represented by the model. Addressing these processes could be prioritized in further iterations of models to facilitate large increases in model accuracy.



Global structures emerge whereby the Shapley value approach can identify thresholds at which features become influential (Fig. 4). Because our approach considers all sites simultaneously, we can make conclusions about the spatial coherence of influential attributes across regions (Mai et al., 2022). A few variables, most prominently soil water content, are highly influential regardless of the value of the variable (i.e., whether small or large). However, some variables largely have no influence until certain thresholds are crossed, such as snow cover, road density, irrigation area, and lake area. The ability to resolve threshold behavior in model performance allows for better parameterization of models and identification of areas where increased data collection could improve model calibration (Zehe and Sivapalan, 2009).

This model diagnostic approach provided intuitive results that match the general understanding of streamflow controls across ecoregions (Figs. 6 and. S5). The features that commonly decreased model accuracy the most at individual sites (min $\psi$) were related to soils & geology, hydrology, and climate predictor groups (Fig. 5). The influence of other predictor groups is more variable. For example, urban features (urban extent, road density, population count and density, and human footprint index) are influential in catchments near large metropolitan areas, such as near Chicago, New York, and Boston, but their influence is largely absent elsewhere. Urban features are the most influential predictors for just 7.7% of all gages, but these urban-controlled sites have low KGE values similar to KGE at sites controlled by the most influential variable group, soils and geology (Fig. 6). In this way, Shapley values show utility in interrogating process-based models by allowing for the identification overarching controls across all sites in a dataset while not obscuring unique, local controls.

## 4.2 Natural and anthropogenic process representation within the NWM

### 4.2.1 Climate

Climate processes are of central importance to the goodness-of-fit for the NWM for many sites (Fig. 5), as indicated by large absolute Shapley values ($\overline{|\psi|}$) for climate variables. These results align with results of multiple studies focused on climate processes as drivers for streamflow processes, such as non-perennial streamflow (Hammond et al., 2021; Price et al., 2021; Zipper et al., 2021) and peak streamflow (McMillan et al., 2018). Shapley values results show that climate processes that are related to low water availability (i.e., low values of precipitation, aridity, and ET) decrease the predictive capacity of the NWM (Fig. 4). The inverse is also true, in that streamflow can be simulated more accurately at sites with higher precipitation and lower ET (Fig. 5). While prior studies have observed the poor performance of the NWM to aridity (Johnson et al., 2023b), fewer have highlighted the potential impact of climate variables to improve model performance at humid locations.

Soil water content, actual ET, and precipitation are the most influential values for determining KGE, all of which are highly seasonal (Elnashar et al., 2021). For example, the spatial map of KGE performance (Fig. 5) is broadly related to precipitation amount and the Shapley value for precipitation (Fig. 3; Lute and Luce, 2017). In areas where climate may have a lower degree of variance throughout the year, NWM accurately simulates streamflow because of the predictability of the hydrologic response in a basin. As an example, we find that the presence of a considerable snow cover (> 40%; Fig. 4) can





improve model predictability, which has been noted elsewhere (Johnson et al., 2023b) and may be related to the predictability of seasonal snowmelt, which can dominate the water balance in cold regions. These results highlight the ability of Shapley values to elucidate the relationships between climate and streamflow and provide important insights into careful parameterization of climate forcings to increase model accuracy.

### 4.2.2 Hydrology

Of the variables in the hydrology category, we observed the largest effect on KGE in the NWM from lake area and upstream reservoir storage relative to annual flow volume (the degree of regulation), with KGE decreasing as lake area and the degree of regulation increase (Figs. 3 and 4). The modeling of pond and lake storage and release is a known deficiency in large-scale hydrologic modeling, and recent parameterizations have been developed to enhance representation of surface-water depression storage (Costigan and Daniels, 2012; Hay et al., 2018; Hodgkins et al., 2024).

The negative impact of lake and reservoir features on model accuracy is greater to the NHM (Fig. S3) than to the NWM (Fig. 4). As noted earlier, the NHM framework does not simulate any kind of reservoir operations, water withdrawals, or stream releases (Regan et al., 2019). On the other hand, the NWM framework models passive reservoir routing (Cosgrove et al., 2024) to mitigate the confounding effects of lake and reservoir volume on model performance. The successful identification of a hydrologic model sensitivity by the Shapley value approach underscores that the method is highly 315 interpretable and can produce intuitive results that match our conceptual models.

### 4.2.3 Physiography (Topography, Soils, and Geology)

Hydrologic connectivity controls many facets of the natural flow regime and determines the ability of a watershed to store and release water (Husic and Michalek, 2022). Parameterizations of soils, geology, and other basin characteristics are highly heterogeneous and mediate many facets of connectivity, many of which are poorly resolved in large-scale 320 hydrologic models (Li et al., 2023). Soil water content was the most impactful predictor for KGE according to the Shapley value analysis, with low values of soil water content greatly impacting the KGE (Fig. 4). Soil water content represents the annual soil water available for evapotranspiration, with complete soil saturation as an upper limit. Other factors that contribute to a high degree of hydrologic connectivity, such as high percent sand and low percent clay (Fig. 5), also highlight the inability of the NWM to resolve storage and connectivity, which likely results from inadequate parameterization of areas 325 that have highly seasonal soil water content (Hughes et al., 2024) and the inability of the current generation of NWM to represent losing streams (Jachens et al., 2021; Lahmers et al., 2021).

We also identified predictor variables commonly associated with the physiography of headwater systems as important predictors of KGE (Fig. 5), such as drainage area and mean elevation. Headwater systems are defined as "surface-water catchment areas and groundwater zones that contribute water, material, and energy to a headwater stream" 330 (Brinkerhoff et al., 2024; Golden et al., 2024). Headwater streams typically have smaller drainage areas and higher mean elevations, which our approach found were associated with lower KGE values for NWM predictions possibly because NWM





simulates fluxes on a 1×1 km$^2$ grid cell and can misrepresent processes that are on the scale of headwater systems. These headwater systems are low-order and highly variable in their flow regimes (Rojas et al., 2020), both of which are inadequately represented in NWM.

**4.2.4 Anthropogenic processes**

Of the variables related to anthropogenic influence, we note that urban features, such as urban extent, road density, population county, population density, and human footprint, typically decrease KGE values for modeled streamflows (Figs. 5 and S4). The construction of urban drainage networks has been recognized to increase the connectivity of water, solutes, and sediment, and to add additional pathways of transport through the artificial routing of water (Lakoba et al., 2020; Zarnaghsh and Husic, 2021). In a continental-scale analysis of the NWM, urban areas exhibited some of the largest bias (Johnson et al., 2023b), in part due to the presence of constructed drainage networks. Notwithstanding this limitation, the NWM has shown some success in simulating hydrology when artificial urban channels, which differ from natural flow paths, are manually delineated within the flow grid (Pasquier et al., 2022). However, manual delineation is not feasible for applications at intended for regional or continental scales, such as NWM and NHM.

Our model identifies a threshold of around 5 km/km$^2$ of roadways as the initiation point whereby the presence of roadways decreases accuracies of NWM and NHM (Figs. 4 and S3). The sensitivity of the roadway density feature may indicate other associated infrastructure, the configuration of proximal impervious areas, and the relative amount of human alternation of surface flow generation and routing mechanisms not picked up by considering imperious area alone. Population and population density similarly likely indicate associated infrastructure that alters flow timing and magnitude of water delivery to rivers (Hopkins et al., 2019). For example, leaky infrastructure can result in elevated low flows beyond natural background levels (Bhaskar et al., 2020). Regarding agriculture, irrigation return flows have been shown to be important to flow generation processes, particularly in lower elevation, arid rivers (Putman et al., 2024). These urban and agricultural features can decrease model accuracy when present, but the absence of these features does not necessarily increase model accuracy (Fig. 5).

**4.3 Limitations and Future Research**

Our interpretable modeling approach has provided several insights into interrogating process deficiencies in the NWM and NHM. Although the inferences we derived from the Shapley values are robust, interpretable, and intuitive, the analysis approach itself is not causative (Lundberg et al., 2020). Thus, some inferences may occur due to indirect correlation (Heskes et al., 2020). We took precautions to mitigate the effect of feature correlations while constructing the random forest model, such as through random exclusion of features during tree construction and out-of-bag sampling (Fox et al., 2017). Our approach provides us with confidence because, as we noted earlier, many of the inferences we derived with the Shapley values match the causative "under the hood" model assessments performed by others (Hodgkins et al., 2024; Hughes et al., 2024; Jachens et al., 2021; Pasquier et al., 2022).



The interpretable modeling approach has its own set of limitations. First, predictions made by Shapley values are a
365 function of (1) the set of sites considered, in this case 4,383 streamgages in the United States used in NWM and NHM
assessment and (2) the choice and performance of the predictive model, which in this case was a reasonably accurate random
forest model ($R^2 \geq 0.43$). With regard to the first point, if our analysis approach were applied to interpreting the KGE values
for streamflow predictions made by applying the Soil Water and Assessment Tool (SWAT) to Europe (Abbaspour et al.,
2015), the order and magnitude of influence by various features would undoubtedly change. To the second point, although
our random forest model is reasonably accurate, it only explains 47% of the variance in KGE prediction for the NWM (and
43% for the NHM). Thus, while our model captures dominant global trends and local structures, more than half of the
variance in predicting the KGE is unaccounted for; ways to further reduce model variances could be explored in future
studies. Further, we only consider the KGE goodness-of-fit metric in this study, but if we were to interpret other goodness-
of-fit metrics, such as the Nash-Sutcliffe Efficiency, there is potential that inferred controls on model performance may
change. This is because all goodness-of-fit metrics encode for – and are biased by – various information contained within
streamflow timeseries (Clark et al., 2021). Nonetheless, of the common evaluation metrics presently applied in the
hydrologic literature, use of the KGE is increasing because of its lower overall bias and provision for balanced results during
low- and high-flow conditions (Althoff and Rodrigues, 2021).

Several limitations and opportunities for improvement exist regarding the data inputs and model outputs. First, the
380 spatial extent and resolution of the catchment attribute dataset may be too coarse, particularly for smaller basins. Of the 48
catchment attributes derived from the BasinATLAS dataset (Linke et al., 2019), spatial resolution can range from 3 arc-
second for elevation to 5 arc-minute for land use (e.g., cropland and urban extents). At 40º N, the median latitude of the
CONUS, these arc values correspond to ~85 meters and ~7 kilometers, respectively. These datasets were aggregated to 15-
arcseconds (~350 m), thus the calculated attributes for smaller basins are more uncertain due to a smaller sample size of
385 attribute estimates within basin bounds. A second data limitation is that the catchment attribute dataset represents snapshot-
in-time value for all basins (Linke et al., 2019). However, catchments and their characteristics, particularly land use, may
change substantially over time. The hydrologic models are simulated over multiple decades (1984 to 2016), during which
change may occur and be captured within the process-based representation of the models but not in the catchment attribute
dataset. Improved spatial resolution and temporal evolution of catchment attributes could provide deeper insights into
390 identifying NWM and NHM process deficiencies. Finally, the process-based models used here vary in their spatial and
physical representation of hydrologic processes. Process-based model differences in routing schema, spatial groupings
(hydrologic response unit vs grid-based), and subsurface properties could result in slight differences but are unlikely to
impact or explain broad, CONUS scale patterns observed in our analysis.

Looking forward, the National Oceanic and Atmospheric Administration (NOAA), the developers of NWM, are
395 expanding modeling capacity with their Next Generation Water Resources Modeling Framework (NextGen; Ogden et al.,
2021). In addition to a uniformly applied national hydrologic model, there will be tools for identifying the best
model/parameterization for each individual location and then modeling regions as patchworks of individual/local models



(Cosgrove et al., 2024). In addition to assessing overall flow performance, this approach could be used for specific components of the flow regime, such as high and low flows. For example, studies that have focused on individual components of non-perennial drying regimes have used a random forest approach coupled with partial-dependency analysis (e.g., Price et al., 2021). The Shapley value approach used in this study could be used in a similar way to evaluate magnitude and directionality of impact between predictor values and flow regimes across systems. Further, modules are planned for purely data-driven approaches, like Long-Short Term Memory models (Frame et al., 2021). Our interpretable modeling approach provides a starting point to inform the parametrization of local-scale and regional-scale applications in the next generation of hydrologic models.

## 5 Conclusions

The interpretable machine learning technique we present is flexible and model agnostic. We use the technique to identify potential process-based deficiencies in two continental scale hydrologic models: NWM and NHM. Compared to other parameter-based continental-scale sensitivity analyses, our approach provides a post-hoc assessment of model sensitivity. This method allows for the identification of thresholds after which a feature begins to negatively impact streamflow model performance. Globally, soil water content was the most common feature influencing the accuracies of streamflow simulations, followed by aridity, evapotranspiration, and precipitation. We interpret the results to indicate that the present formulations of NWM and NHM have limited ability to resolve soil water storage and release, particularly in arid locations. Locally, the presence of lakes and flow regulation were related to decreased model accuracy as were roadways and irrigation canals. Our results suggest that further refining how these influential processes are represented in large scale hydrological models would result in the largest increases in model accuracies. This study demonstrates the utility of interrogating process-based models using data-driven techniques and explainable AI, which has broad applicability and potential for improving simulation of large-scale hydrology and water quality.

## Code availability

The data and code used for the random forest model, the Shapley value analysis, and generation of figures can be found at the following Open Science Framework link: https://doi.org/10.17605/OSF.IO/MNQCZ.



## Author contributions

**AH**: Conceptualization, Methodology, Formal analysis, Visualization, Writing – original draft, Writing – review & editing.
**JH**: Methodology, Formal analysis, Writing – original draft, Writing – review & editing. **AP**: Methodology, Writing – original draft, Writing – review & editing. **JK**: Writing – original draft, Writing – review & editing.

## Competing interests

The contact author has declared that none of the authors have any competing interests.

## Acknowledgements

This work was supported by the National Science Foundation (Award Nos. 2229616 and 2438017). This work was performed at the HPC facilities operated by the Center for Research Computing at the University of Kansas supported in part through the National Science Foundation MRI Award OAC-2117449. Any use of trade, firm, or product names is for descriptive purposes only and does not imply endorsement by the U.S. Government. This work has been reviewed by the U.S. Forest Service. This product has been peer reviewed and approved for publication consistent with USGS Fundamental 435 Science Practices (https://pubs.usgs.gov/circ/1367/).



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
