# Peer review of "Interrogating process deficiencies in large-scale hydrologic models with interpretable machine learning"

_EGUsphere, 2024_

## Author Response (AR1)

**Authors' Response to Review Comments:**

The authors would like to thank Jonathan Frame, an anonymous reviewer, and the associate editor for their time and constructive comments regarding our manuscript. We have acknowledged the hard work of the editor and reviewers in the Acknowledgements section of the revised manuscript.

In this document, the authors list the major comments made during the review of the original manuscript and, underneath each comment, **our responses are written in bold font**.

**Reviewer #2 Comments:**

This manuscript uses Shapely approach, an explainable AI methodology, to broadly define categories that contribute to model bias, with the focus being on streamflow output from two, continental-scale, processed based, hydrological models. The methodology uses a random forest model to predict KGE for streamflow in each model and it is trained on several ecoregion characteristics. The Shapely values indicate feature importance for all the ecoregion characteristics and their impact on streamflow KGE. The random forest model is moderately sufficient in predicting KGE. This study, rather than assessing model performance or specific ways to improve model performance, is a proof of concept for using explainable AI for process-based hydrological model bias identification sources in a post-hoc manner.

**We thank the reviewer for the thorough reading and critique of the manuscript. Their comments have helped improve the manuscript's quality. In particular, we agree that the study can be framed more clearly as a proof-of-concept and we now emphasize that throughout the manuscript. Our specific changes to this end are noted under individual reviewer comments below. New sentence in the abstract is below:**

> *Large-scale hydrologic models are increasingly being developed for operational use in the forecasting and planning of water resources. However, the predictive strength of such models depends on how well they resolve various functions of catchment hydrology, which are influenced by gradients in climate, topography, soils, and land use. Most assessment of hydrologic model uncertainty has been limited to traditional statistical methods.* ***Here, we present a proof-of-concept approach that uses interpretable machine learning techniques to provide post-hoc assessment of model sensitivity and process deficiency in large-scale hydrologic models.*** *We train a random forest model to predict …*

Overall, this manuscript is well written and organized. There were few grammatical errors, and the sections of the manuscript are logically structured. This methodology is an interesting and scientifically significant way to assess sources of model bias and parameter sensitivity for continental-scale models, which are typically far too computationally expensive for traditional sensitivity analyses. I suggest the background and methodology can be expanded in some sections for increased clarity, particularly across scientific fields. Additionally, the authors should clarify specifically the purpose and motivation. The main points of revision are as follows:

**We respond to each of the reviewer's main points below:**

1. Purpose and Motivation:

    1. Clearly define the study's purpose: This is not a model comparison or a process representation assessment. This reads more as a proof of concept for a new methodology to assess bias and sensitivity in process-based, continental-scale hydrologic models.

        **We now set up the limitations of traditional sensitivity analyses in the Introduction section, namely their computational intractability.**

        *The sensitivity of process-based hydrologic models to certain catchment attributes and parameters has been interrogated using well-established statistical tools, such as sensitivity analysis (Pianosi et al., 2016; Song et al., 2015). These approaches work by exploring the range of values that model parameters may take and recording the net impact on model performance (Mai, 2023). Notable examples include the Sobol' (2001) and Morris (1991) methods. **A drawback of traditional sensitivity analysis methods, particularly when applied to large-scale hydrologic applications (Mai et al., 2022), is that they can be computationally demanding (Sarrazin et al., 2016). Thus, there is a need to develop alternate methods for assessing model sensitivity that are useful in scenarios where traditional methods are computationally intractable.***

        **We follow up the discussion of present-day limitations that leads us into the study's purpose which we put into the objective paragraph:**

        *This paper aims to interrogate large-scale hydrologic model performance with machine learning tools to identify which processes may be inadequately represented in physically based models. Thus, the questions we address are: what catchment attributes can be used to predict poor model performance, and are certain dominant hydrological processes associated with these catchment attributes? To **answer these questions, we present a proof-of-concept approach that uses machine learning techniques to provide post-hoc assessment of model sensitivity …***

    2. Strengthen the introduction and motivation by emphasizing the challenges of running traditional sensitivity analyses on computationally expensive, large-scale models. This should serve as a primary justification for developing and applying this methodology.

        **We have rewritten the 3ʳᵈ and 4ᵗʰ paragraphs to clearly establish the existing limitations and make it explicit how our approach helps to close some existing methodological and knowledge gaps.**

        *The sensitivity of process-based hydrologic models to certain catchment attributes and parameters has been interrogated using well-established statistical tools, such as sensitivity analysis (Pianosi et al., 2016; Song et al.,*

*2015). These approaches work by exploring the range of values that model parameters may take and recording the net impact on model performance (Mai, 2023). Notable examples include the Sobol' (2001) and Morris (1991) methods. **A drawback of traditional sensitivity analysis methods, particularly when applied to large-scale hydrologic applications (Mai et al., 2022), is that they can be computationally demanding (Sarrazin et al., 2016). Thus, there is a need to develop alternate methods for assessing model sensitivity that are useful in scenarios where traditional methods are computationally intractable.***

*Explainable or interpretable machine learning methods have the potential to bridge the gap between data-driven insights (provided by machine learning models) and process-based understanding (contained within physically based models) (Slater et al., 2025). Several explainable machine learning methods have been developed, including Partial Dependence Plots (PDP; Friedman, 2001), Local Interpretable Model-Agnostic Explainers (LIME; Ribeiro et al., 2016), and Shapley Additive Explanations (SHAP; Lundberg et al., 2020). **In hydrology, for example, these tools have been applied for the analysis of soil moisture (Huang et al., 2023), water table depth (Ma et al., 2024), and drought intensity (De Meester and Willems, 2024). Interpretable machine learning can complement and enhance traditional sensitivity approaches (Maier et al., 2024), by providing post-hoc interpretative insights into how parameter changes influence hydrologic model predictions, that is, without the need for perturbing the model parameter space.** Interpretable machine learning methods are not without limitations as they only imply relations in the model which may not necessarily be causal (Heskes et al., 2020), thus caution should be exercised when interpreting results.*

2. Improved Model Description:

   1. Include a more detailed description of model configurations and processes of interest (even in supplementary materials, if necessary).

      **We are not sure that we completely understand this request, but we have improved Section 2 Methods to be more easily understood based on the reviewer's specific comments below.**

   2. Clearly identify the configurations used and ensure consistency across the text (e.g., clarify references to model attributes in results and Line 390).

      **The reference to spatial and temporal catchment attributes is in regard to the resolution of the 48 predictors appearing in Figures 3 and 5.**

   3. Focus on processes directly relevant to the discussion (e.g., those affecting streamflow).

      **All processes discussed in the article are now relevant to streamflow generation and estimation.**

3. Clarify Objectives and Takeaways:

   1. One suggestion is to change the recurring message: the Shapley results are a tool for identifying model deficiencies and offering insights into bias and process representation improvements, not for directly improving NWM or NHM processes. This study is a proof of concept for using of explainable AI for hydrologic model bias identification in a post-hoc manner and authors can emphasize that this study demonstrates the utility of explainable AI in detecting model deficiencies, particularly for computationally expensive, large-scale hydrological models. Authors can compare the methodology to traditional sensitivity analyses, highlighting its innovation and feasibility given the computational constraints of large-scale models.

      **This is a great suggestion by the reviewer and we agree that the paper can be more effectively framed as a proof-of-concept tool for identifying model deficiency and not for directly improving NWM or NHM. We now spend more time in both the Abstract, Introduction, Objective Paragraph, and Discussion relating our method to traditional sensitivity analysis approaches.**

Specific Comments:

1. Line 30: "Grand challenge of hydrology…" Why/what makes this especially difficult at large scales? What are the specific challenges for large scale models that are addressed with this methodology?

   **We note several reasons why this is difficult at large scales, including (1) equifinality, (2) scaling, (3) epistemic uncertainties, and (4) spatial heterogeneity in processes. Our methodology most directly addresses point (4) regarding spatial heterogeneities in watershed processes and how these impact streamflow generation.**

2. It is not initially clear why the analysis includes both the NWM and NHM. Justification for this should be added. Does using two models illustrate that this methodology is useful beyond a single-model use case? Or something else?

   **We apply the method to two models to see if the method can identify unique deficiencies that are a function of the model structures. In the end, yes – our method achieves this. The method identifies that reservoir water storage negatively impacts flow prediction more so in the NHM (which does not consider reservoir storage) than in the NWM (which does explicitly consider reservoir storage). We make it clear in the intro why we consider multiple hydrologic models.**

   **In introduction, we add the following:**

*Perhaps the major difference in the two modeling approaches is that the NWM has a focus on high-resolution (hourly) flood forecasting whereas the NHM is designed to assess general water availability at timescales from days to centuries (Towler et al., 2023). The NWM and NHM have variable success for streamflow prediction (Tijerina et al., 2021), with the strength of prediction varying as a function of catchment-scale climate, land use, and physiography.* **As the NWM and NHM have different constructions, their sensitivity to catchment drivers is likely to differ.**

3. Generally, the use of ecoregions needs to be expanded upon in the introduction and methods. It should be a bigger part of the central thesis, since the analysis and discussion consist primarily of the catchment attributes of these ecoregions. Additionally, it is not entirely clear in the methodology how streamflow gages are related to the ecoregions and catchment attributes. What is the catchment size / product (e.g., NHDPlus)?

   **The ecoregions are used solely to see if the model sensitivities (identified by explainable AI) are a function of spatial variation in the US. A single model of KGE for all sites was trained, rather than a model for each ecoregion.**

   **Streamflow gages and their respective drainage areas were assigned to the ecoregion they fell within. The catchment attributes for each drainage area were calculated from BasinATLAS (Linke et al., 2009). Because ecoregions represent spatial variation, the catchment attributes of streamgages in the same ecoregion will be relatively similar while the attributes of streamgages in differing ecoregions will differ more greatly.**

   **We add more justification of ecoregions to the methods.**

   > **We used the Ecological Regions of North America as a way of grouping clusters of catchments in order to facilitate the discussion of similarities (or dissimilarities) between the drivers of model performance across broad areas (Omernik, 1987)**. Ecoregions are defined by "perceived patterns of a combination of causal and integrative factors including land use, land-surface form, potential natural vegetation, and soils" (Omernik, 1987). **Results from individual catchments were aggregated to the ecoregion level for comparison of general trends**. A catchment was assigned to an ecoregion based on the greatest area of an ecoregion contained within the drainage boundary of a catchment.

4. Line 20 and 71: "model performance" – should indicate that streamflow performance is the only variable being assessed.

   **We have clarified that streamflow is being assessed in the abstract.**

5. Line 26: *Why* are large-scale hydrologic models important? Should add brief justification of the rational of using these vs. e.g. regional, catchment scale models.

   **We note justifications for the use of large-scale models over smaller-scale models, including their comprehensive spatial coverage,**

**consistency/transferability in methodology across regions, and their ability to inform transboundary water policy.**

6. Line 29: Besides parameterizing and calibrating, what about physics and physical process representation in these models?

   **This is an important question, but beyond the capabilities of our approach or this study.**

7. Line 44: Clarify what is meant by "sites".

   **A "site" is a streamgage and its upstream drainage.**

8. Line 54: Can also cite Ma et al., 2023 (Groundwater) https://doi.org/10.1111/gwat.13362

   **Thank you for the reference. We now cite this study. We moved sentences around so the citation is later on in the paragraph.**

9. Lines 54-55: This is too broad a statement. Needs further explanation or an example given.

   **This statement has been deleted as the section was rewritten.**

10. Line 60: Have numerous explainable AI methods been developed for use in hydrology or just generally? More generally, expand on what explainable AI is and how it is defined. Authors discussed that XAI can be leveraged and cite methods, but there is not a clear explanation of what XAI is.

    **We cite three studies are now cited that have been used in hydrology:**

    *In hydrology, for example, these tools have been applied for the analysis of soil moisture (Huang et al., 2023), water table depth (Ma et al., 2024), and drought intensity (De Meester and Willems, 2024). Interpretable machine learning can complement and enhance traditional sensitivity approaches (Maier et al., 2024), by providing post-hoc interpretative insights into how parameter changes influence hydrologic model predictions, that is, without the need for perturbing the model parameter space.*

11. Line 90: Clarify if these basins are NHDPlus or something else.

    **Yes, as these are USGS gages they are.**

12. Line 92-93: This seems more like a result and might fit better in the results section.

    **The NHM and NWM model outputs were not produced by us, rather we just use the products of other researchers, thus we keep this line in the methods.**

13. Line 94: separate the metrics section from the model section as these are not explicitly related. Also, provide justification for only using KGE.

    **We separate these sections. Further, justification for the KGE and discussion of alternate metrics is listed in the Discussion.**

14. Line 105: It is not clear what the "aggregated" attributes are. Are these the 7 groupings listed in the next sentence?

**We remove the term "aggregated" and replace it with "selected". Yes, those are the seven groupings.**

15. Line 110-111: Need to clarify where soil water content is being "represented." In the models or elsewhere?

**This sentence has been corrected. The authors of this study are using the products in BasinATLAS, thus "we" are not representing the value. Rather, the soil water content in the BasinATLAS dataset is calculated as described and we simply use this value as a catchment attribute.**

16. Line 112: The "in-the-bag" and "out-of-bag" language is jargon and can be explained in plain language. The language could be revised to be more consistent with the "training-testing" language in the results. Further explanation can be provided on the RF methodology, particularly because this paper is outlining a new methodology that others will likely want to employ and will need more specifics to do that. E.g., what was the training-testing split as a percentage? Was this a randomized selection?

**We keep the in-the-bag and out-of-bag terminology as it is true to the context our modeling approach. We do bootstrapping rather than cross-validation so it is more appropriate to write it the way we have. We use bootstrapping in lieu of cross-validation, thus the percentage (e.g., 80:20) is not explicitly defined but depends on the amount of random sampling. In our study, this split tends towards 66:34 (training-testing) but varies for each tree in the random forest.**

17. Line 125: Unless someone is familiar with Shapely values and methodology, some of the results are initially difficult to interpret. Consider adding some general explanation of what feature importance, Shapely value, and directionality mean within the context of the results and figures. Also, it would be helpful to consider explaining further in the methods / changing the language about how KGE behaves (perhaps instead of increase/decrease use improve/worsen) so that the qualitative relationship between Shapely and KGE is clear.

**In earlier iterations of this manuscript, we used terminology such as "improve/worsen" but this can be viewed as subjective language, thus we use more the more exact language of "increase/decrease. This was done to comply with USGS style guidance.**

**Nonetheless, the point is taken regarding the complexity of the topic. We add a new paragraph to the method to describe a more plain language summary of how Shapley values work:**

*To aid in interpretation of Shapley values, we provide a brief example. The random forest model described in Section 2.2 is trained to predict the KGE of the NWM (or NHM) model at 4,383 sites in the analysis. In short, "how accurate is the NWM model at site?" The random forest model transforms 48 catchment attribute features into a prediction of KGE. In the absence of Shapley values or interpretable AI, the process by which the*

*catchment attributes are transformed to create the KGE prediction is uncertain, that is, it is a blackbox. Shapley values elucidate the marginal contribution of a feature ($\psi$) to the random forest, which is defined as how much the predicted KGE at a site increases ($+\psi$) or decreases ($-\psi$) when a feature is included in the model. In this way, sensitive features will have a high magnitude of Shapley values, $|\psi|$, as the predicted KGE is sensitive to the value that the feature takes. Thus, Shapley values help to distinguish the catchment attributes that cause variation in predicted KGE across space. Although the full range of Shapley values for the 48 catchment attribute features are informative, we focus on the most impactful feature negatively affecting model performances at each site. The most impactful feature is defined as the one having the lowest Shapley value (min $\psi$) at a site, indicating the most negative marginal contribution to KGE prediction, that is, reducing the predicted accuracy of the NWM (or NHM)*

18. Figure 5: Explain difference between Shapely value and the Predictor Value, this will help in interpreting the swarm plot.

    **The Predictor Value is the value of the explanatory variable (e.g., Precipitation). Thus, when the markers in the swarm plot are dark, the precipitation is low at that basin relative to other basins. On the other hand, when the markers in the swarm plot are bright, the precipitation at that basin is high relative to other basins.**

    **We add this explanation to the caption.**

19. Line 128: Expand on this point and give reasoning for why this is beneficial within this methodology. Explain "distribution of gain."

    **Detailed description of the four properties is beyond the necessity of the text and the reader is left to follow up on the in text citation to read more.**

20. Line 146: Consider moving the ecoregion names listed here to the figure caption and replace with content focused on why this methodology was chosen, the implications of this method on the study, general ecoregion methodology explanation. (See Comment 6).

    **This is now done.**

21. Line 154: A plot of actual to modeled KGE would be helpful (maybe in the Supplementary Materials). Did authors evaluate this with any other metrics?

    **This is now shown in the paper and below for the reviewer. We briefly looked at NSE, but it did not show much of a different story than KGE.**

[Figure]

**Figure 3.** Random forest model evaluation for KGE prediction. The binned scatter plot displays the count of data points in each partitioned bin. For visual clarity, predicted and observed KGE values less than 0 are not plotted, although they are included in the calculation of $R^2$ for each model.

22. Line 249-242: This is a great explanation. Perhaps include something similar in the introduction as this is an important point which makes this study unique. Remove the second "that is."

    **Thank you – we now include this statement in the introduction to justify the work. We also remove the second "that is".**

23. Line 272: after "certain thresholds are crossed" reference specifically the scatter plots in Fig. 4.

    **The figure is now referenced.**

24. Section 4.2 Title: Rethink the section heading. The section broadly discusses model performance related to the ecoregion features, not model formulations or actual process representations.

    **We change the section heading to "Natural and anthropogenic drivers of NWM and NHM performance" to be more consistent with the model insights.**

25. Line 332: The NWM routes streamflow through the NHDPlus vector network, not across a 1km$^2$ The reasoning for headwater performance should be given more thought.

    **The flow is routed as you say, but the landscape fluxes are produced at the resolution we describe.**

26. Line 362: "under the hood" model assessments – this is jargon and can be clarified with plain language.

    **We change this term to "causative and mechanistic model assessments performed by others".**

27. Figure 3: Specify on the x-axis this is *predicted*.

    **This is given in the figure caption.**

28. Figure 4, Figure S2, Figure S3: Need colorbar explanations.

The colorbars represent the values of the Shapley value ($\psi$) for each panel. This has now been noted.

---

## Author Response (AR2)

Nearly all comments have been carefully addressed. It is good that it is now clear from the introduction that this study provides a proof of concept, several comments from the reviewer have herewith been addressed.

1. Please extend your explanation of why you focus on 2 instead of 1 models. It is perfectly fine and even interesting, but the added sentence ' As the NWM and NHM have different constructions, their sensitivity to catchment drivers is likely to differ' does not provide sufficient explanation for a reader who does not know the two models and the most important differences between them. The response to reviewers is more extended and valuable.

**Apologies for the oversight, we have now rewritten this intro paragraph and included what exactly differs and why these differences are crucial for our approach. Key notes and changes are underlined.**

> *"The National Water Model (NWM) and the National Hydrologic Model (NHM) are two process-oriented, continental-scale hydrologic models used in operational decision-making (Towler et al., 2023). The NWM framework applies the Weather Research and Forecasting Hydrologic model (WRF-Hydro) formulation, which simulates infiltration, evaporation, transpiration, overland flow, shallow subsurface flow, baseflow, channel routing, and passive reservoir routing, but not active reservoir management (Cosgrove et al., 2024). The NHM framework applies the Precipitation-Runoff Modeling System (PRMS) formulation, which represents evaporation, transpiration, runoff, infiltration, interflow, groundwater flow, and channel routing, but not reservoir operations, water withdrawals, or stream releases (Regan et al., 2019). See Text S1 for more details on each model. A key distinction is that the NWM targets high spatial (~250 m) and temporal (hourly) resolution flood forecasting. In contrast, the NHM assesses long-term water availability at hydrologic-response-unit scales (~100 km$^2$, driven by daily forcing) (Towler et al., 2023). Both models exhibit spatially variable streamflow skill across US catchments (Tijerina et al., 2021), with the strength of prediction varying as a function of catchment-scale climate, land use, and physiography. Collectively, differences in resolution, process formulation, and treatment of human regulation make the NWM–NHM pair an ideal testbed for structural sensitivity analysis: drivers influential in both frameworks likely denote overarching hydrologic controls, whereas divergent sensitivities flag processes that are represented differently (or omitted) in either approach."*

2. Reviewer 2 requested more specification on the NWM and NHM in the Methods section or supplementary materials, this has not been addressed, while most readers will not be familiar with the two models on which data this study is centered. I strongly argue that a brief description or overview table of the 2 models is required, this can briefly be done in the main text and extended in the supplementary information. The reader should not be obliged to read the other papers on the models to get to know more details. For the analysis it is also relevant to know more details on the difference between the models than the text that is currently provided in the introduction.

Thank you for bringing this point to our attention, we fully agree that providing more information on the models behind this study would aid the readers in interpreting our results. We have provided greater overview information on the two models in the introduction as described above, and we have added a text section in the supplement providing more detail on both of the models.

We have added this text at the start of the methods section:

*"Text S1 summarizes the models that produced the data used in this study."*

We have added a new supplement (Text S1):

***Text S1: An overview of the National Water Model (NWM) maintained by the Office of Water Prediction of NOAA and the National Hydrological Model (NHM) maintained by the U.S. Geological Survey. These two national-scale models, while both aiming to simulate hydrological processes across the continental U.S. (CONUS), differ significantly in their underlying modeling frameworks, primary operational objectives, spatial discretization, input datasets, and the specific hydrological processes they explicitly represent. Please see Towler et al. (2022) for additional details on each model.***

***National Water Model (NWM) Version 2.1***

*The National Center for Atmospheric Research (NCAR) developed WRF-Hydro, an open-source hydrologic model that serves as the foundation for the National Oceanic and Atmospheric Administration's (NOAA) National Water Model (NWM). NWM simulates and forecasts key water components (e.g., evapotranspiration, snow, soil moisture, streamflow) in real-time across the continental U.S., Hawaii, Puerto Rico, and the U.S. Virgin Islands. NWM version 2.1 utilizes 1 km atmospheric data from NOAA's Analysis of Record for Calibration (AORC) and employs the Noah-MP land surface model to compute energy and water states on a 1 km grid. Hydrologic routing occurs on a 250 m resolution terrain grid, utilizing WRF-Hydro's baseflow parameterization and the Muskingum–Cunge river routing scheme on an adapted NHDPlus version-2 river network. The model features a level-pool scheme for 5,783 lakes and reservoirs, although it lacks active reservoir management. While operational data assimilation is included, it is not applied in the retrospective simulations. Calibration of 14 parameters occurred from water years 2008 to 2013, validated against data from 2014 to 2016 across 1,378 gaged basins.*

***National Hydrological Model (NHM) Version 1.0***

*The U.S. Geological Survey (USGS) developed the National Hydrologic Model (NHM, version 1.0) based on the Precipitation–Runoff Modeling System (PRMS), a modular system often employed for water resource assessment and scenario analysis. NHM simulates water flow and storage processes, including snowpack, soil, and stream networks, using daily discharge simulations. The NHMv1.0 results used here come from a calibration workflow focused on observed streamflow and the Muskingum–Mann routing option. Climate inputs consist of 1 km resolution daily precipitation and temperature data from Daymet. The model's spatial structure is defined by geospatial fabric version 1.0, which for PRMS typically delineates Hydrologic Response Units (HRUs). Calibration employs a stepwise approach to optimize parameters for water budgets and streamflow, first aligning hydrologic responses to baseline observations and then timing streamflow against data from 7,265 headwater watersheds. Final calibration occurs at 1,417 stream gage locations. The calibration period spans odd water years from 1981 to 2010, with*

*validation using even years. NHM does not simulate reservoir operations or water withdrawals; it outputs daily streamflow for analysis.*